# Effect of timing of implementation of containment measures on Covid-19 epidemic. The case of the first wave in Italy

Laura Timelli ⓘ *◎, Enrico Girardi◎

"L. Spallanzani" National Institute for Infection Diseases, IRCCS, Rome, Italy

◎ These authors contributed equally to this work.
* laura.timelli@inmi.it

**Data Availability Statement:** The data underlying this study are available on GitHub (https://github.com/laura2802/Covid19work/find/main).

## Abstract

There is evidence that adoption of non-pharmaceutical containment measures (NPMs) may have had a major impact on Covid-19 epidemic dynamics, and mitigated its effect on health-care system. Optimal timing of implementation of these measures however is not known. In Italy, a national lockdown was decided on March 11[th] 2020 and ended 4[th] of May. At that time, cumulative incidence (CI) was different in Italian regions which ranged from <5 cases/100,000 to >11 cases/100,000 inhabitants. In this paper, we aim to evaluate how level of incidence in different regions at the time of implementation of NPMs affected CI and had an impact on the healthcare system in terms of ICU bed occupancy and mortality rates. We used regional daily new COVID-19 diagnosed cases as well number of people hospitalized in ICU and number of deaths for period February 24-May 11 from all the 19 Italian regions and two autonomous provinces. For each region we calculated: temporal daily trend of cumulative cases of Covid-19/100,000 inhabitants, daily trend of ICU bed occupancy and mortality rate at the end of period. We found that the epidemic curves show similar trends for all regions and all tend to flatten between 11–32 days. However, after 2 months, regions with lower CI at lockdown remained at substantially lower CI (<265 cases/100,000), had a peak of percentage of cases hospitalized in ICU which did not exceed 79.4% and a mortality<0.27/1,000. On the other hand, in regions with higher incidence at lockdown, CI reached 382–921 cases/100,000, the peak of percentage of cases hospitalized in ICU and mortality rate reached 270%, and 1.5/1,000, respectively. Our data suggests that level of CI at the moment of lockdown is important to control the subsequent spread of infection so NPMs should be adopted very early during the course of Covid-19 epidemic, in order to mitigate the impact on the healthcare system and to reduce related mortality.

## Background

From February 2020, Covid-19, the disease caused by the infection with severe acute respiratory syndrome coronavirus 2 (SARS-CoV-2), has rapidly spread across the world [1]. In response to the growing numbers of cases and deaths due to this disease, countries have

**Funding:** This study was funded by the Italian Ministry of Health- Ricerca Corrente INMI Spallanzani- IRCCS – Linea 1, to EG.

**Competing interests:** The authors have declared that no competing interests exist.

implemented measures to control their epidemics, and to preserve healthcare systems. These interventions, also referred to as no pharmacologic measures (NPMs) include: i) personal protective measures (e.g., masks and hand hygiene); ii) environmental measures (e.g., disinfection and ventilation); iii) social distancing measures (e.g., school and workplace closures); and iv) travel related measures (e.g., travel restrictions) [2, 3]. There is evidence that the adoption of NPMs may have had a major impact on Covid-19 epidemic dynamics, slowing the spread of the virus and may have mitigated its effect on the healthcare systems [4–6]. It has been suggested that the timing of implementation of NPMs may have an important effect on the number of cases, and consequently on the burden on the health care system [7]; the data on this issue is however limited.

Italy was the first European country to be severely hit by the Covid-19 epidemic. In February 2020, local SarsCov2 transmission clusters were identified in Northern Italy; in the following month the number of cases rapidly grew in northern Italian regions and the epidemic progressively spread in regions of central and the southern Italy [8]. Nationwide school closure was ordered on March 5[th], public events were banned, social distancing encouraged, self-isolation if ill and quarantine if tested positive on March 9[th]. On March 11[th] 2020 a national lockdown was ordered, meaning closure of all public places and requirements for residents to stay within their home and travel restrictions, this ended the 4[th] of May (Table 1).

The degree of spread of Covid-19 in Italy when NPMs were adopted at national level was extremely heterogeneous by geographical area with Northern Italy being impacted more compared to Central and Southern part [9]. This gives the opportunity to retrospectively evaluate how the different implementation timing affected the evolution of the epidemic in different areas of the country. We, therefore aimed to evaluate how the level of cumulative incidence (CI) in different Italian regions at the time of implementation of mitigation measures, affected the CI and the impact on the healthcare system during the initial phase of the epidemic.

## Methods

Italy is administratively divided in 19 regions and 2 autonomous provinces (PA). Since the 24[th] of February2020, the Italian Ministry of Health, started the daily collection of new Covid-19 cases, where each Region and PA [10] provided the number of deaths, number of hospitalized cases, and number of cases in intensive care unit (ICU). A Covid-19 case was defined as a positive person with microbiologically confirmed SARS Cov-2 evaluated by Rt-PCR on oral swab samples [9].

This study was retrospective and we used regional daily new COVID-19 diagnosed cases as well number of people hospitalized in ICU and number of deaths for the period February

**Table 1. Non-pharmacologic measures for Covid-19 mitigation and dates of issuance in Italy.**

| Measure | | Date | Source |
|---|---|---|---|
| Nationwide school closures | | March 5 | https://www.gazzettaufficiale.it/eli/id/2020/03/01/20A01381/sg |
| Public events banned | | March 9 | https://www.gazzettaufficiale.it/eli/id/2020/03/08/20A01522/sg |
| Social distancing encouraged | A distance of more than 1 meter has to be kept and any other form of alternative aggregation is to be excluded | March 9 | https://www.gazzettaufficiale.it/eli/id/2020/03/08/20A01522/sg |
| Case-based measures | Strong recommendation to stay at home and limit social contacts as much as possible if ill (fever greater than 37.5˚C) and quarantine (absolute prohibition of mobility from one's home o residence) if tested positive | March 9 | https://www.gazzettaufficiale.it/eli/id/2020/03/08/20A01522/sg |
| National Lockdown | Closure of all public places, people have to stay at home except for essential travel | March 11 | https://www.gazzettaufficiale.it/atto/stampa/serie_generale/originario |

24-May 11 from all the 19 Italian regions and the two autonomous provinces (hereinafter all referred to as regions) (NUTS 2) [10]. February 24[th] was the date of starting data collection, May 11[th] was the date one week after the end of the National lockdown. We extended the observation period one week more to reduce bias due to notification delays.

## Statistical analysis

For each region we calculated: 1) the temporal daily trend, on a logarithmic scale of cumulative incidence, i.e., number of cumulative cases of Covid-19 per 100,000 inhabitants [11] reported within each specific date; 2) the daily trend of percentage of ICU bed occupancy, i.e., number of patients in ICU out of the regional bed ICU availability at the beginning of the epidemic (pre Covid-19) [12] and the maximum ICU bed occupancy rate, i.e., maximum number of ICU beds occupied per 100,000 inhabitants; the mortality rate at the end of period, i.e. number of deaths per 1,000 inhabitants.

We also calculated the trend of daily reported new cases of COVID-19 by region and overall Italy using 7-day moving averages to smooth out fluctuations.

Then, for each region, we calculated the number of days (hereinafter referred to as delay) between the date on which each region reached the lowest regional CI (i.e., 1.0 per 100,000 inhabitants at the date of National lockdown, LCI) and the date of National lockdown. To explore the relationship between the CI at the end of the observation period and the delay of declaration of National lockdown, a scatterplot was then visualized pairing the delay and CI at the end of the period for each Region and a correlation coefficient was calculated. In an analogous way, delay was paired with mortality, with the highest percentage of ICU bed occupancy and with the maximum ICU bed occupancy rate reached during the period.

Data was analyzed using Stata version 16.

## Results

At the date of National lockdown, CI was lower than 5/100,000 inhabitants for eight regions, between 5 and 11/100,000 for five and higher than 11/100,000 for ten regions. (Table 2). It was noted that most of the Northern Regions had a high CI and, on the other hand, the Central-Southern regions had a low CI.

As shown in Figs 1 and 2, the incidence of cases was rapidly increasing in all regions, and then flattened in all regions after 21 days, on average, with a minimum of 11 and a maximum of 32 days after the lockdown (Figs 1 and 2). The trends were also characterized by daily fluctuations likely due to variability on the reporting. For this reason they are shown as weekly moving average values.

Compared to the region with the lowest cumulative incidence at the date of lockdown, the median delay in implementation of NPIs was 2 days (IQR: 2–4) in the 7 regions with low CI, 7 days (IQR: 7–9) in regions with intermediate CI and 15 days (IQR: 14.5–16) in regions with high CI.

Fig 3, shows a scatterplot by Region of the relationship of CI two months after the date of lockdown (i.e. May 11[th]) and the delay. Overall, regions with shorter delay remained at substantial lower incidence during the considered period compared to regions with longer delay, and we observe CI increased as delay was increasing with a correlation of 0.64 (95%CI 0.41–0.87).

We analyzed trends of admission in ICU by region expressed as percentage of ICU beds available before the start of the pandemic (Fig 4). At implementation of lockdown, this percentage was below 100% in all regions, thereafter, however, in seven regions it would have

**Table 2. Cumulative Incidence (CI) /100,000 of COVID-19 at date of National lockdown (March 11th, 2020) and after 2 months (May 11th, 2020), highest percentage Intensive Care Unit (ICU) bed occupancy, maximum ICU bed occupancy rate and mortality rate (May 11th, 2020) in Italy.**

| Region (NUTS 2) | Geographical area | Delay (days) | CI at March 11th (per 100,000) | CI at May 11th (per 100,000) | Highest percentage bed ICU occupancy | Maximum ICU bed occupancy rate | Mortality rate (per 1,000) |
|---|---|---|---|---|---|---|---|
| Calabria | South | 0 | 0.98 | 58.24 | 15.75 | 1.18 | 0.05 |
| Basilicata | South | 2 | 1.42 | 68.58 | 38.78 | 3.38 | 0.05 |
| Puglia | South | 3 | 1.91 | 107.40 | 52.30 | 3.95 | 0.11 |
| Sardegna | South | 3 | 2.26 | 81.91 | 23.13 | 1.89 | 0.07 |
| Abruzzo | South | 4 | 2.90 | 236.89 | 61.79 | 5.79 | 0.28 |
| Sicilia | South | 4 | 1.66 | 66.78 | 19.14 | 1.60 | 0.05 |
| Campania | South | 5 | 2.65 | 79.32 | 54.03 | 3.12 | 0.07 |
| Lazio | Center | 5 | 2.55 | 122.30 | 35.55 | 3.45 | 0.10 |
| P.A. Bolzano | North | 5 | 14.12 | 484.21 | 175.68 | 12.24 | 0.55 |
| Molise | South | 7 | 5.24 | 125.32 | 30.00 | 2.94 | 0.07 |
| P.A. Trento | North | 7 | 14.23 | 794.13 | 253.13 | 14.97 | 0.82 |
| Valle d'Aosta | North | 7 | 15.92 | 921.49 | 270.00 | 21.49 | 1.11 |
| Toscana | Center | 8 | 8.58 | 262.41 | 79.41 | 7.96 | 0.25 |
| Umbria | Center | 8 | 5.22 | 160.09 | 68.57 | 5.44 | 0.08 |
| Friuli Venezia Giulia | North | 9 | 10.37 | 258.22 | 50.83 | 5.02 | 0.26 |
| Marche | Center | 11 | 31.40 | 428.97 | 146.96 | 11.08 | 0.63 |
| Piemonte | North | 11 | 11.50 | 660.54 | 138.53 | 10.40 | 0.78 |
| Liguria | North | 14 | 12.51 | 569.57 | 99.44 | 11.54 | 0.83 |
| Emilia-Romagna | North | 15 | 39.00 | 602.67 | 83.52 | 8.41 | 0.87 |
| Veneto | North | 15 | 20.85 | 382.01 | 108.87 | 7.26 | 0.34 |
| Lombardia | North | 17 | 72.36 | 813.78 | 160.39 | 13.73 | 1.50 |

Delay are days between the date on which each the region reached the cumulative incidence equal to the lowest regional CI at time of lockdown and the date of lockdown; highest percentage bed ICU occupancy is the maximum number of patients in ICU out of the regional bed ICU availability (before Covid-19) in the reference period; maximum ICU bed occupancy rate is the maximum number of ICU beds occupied per 100,000 inhabitants; mortality rate is expressed as number of Covid-19 deaths per 1,000 inhabitants. Regions are reported by ascending order of delay.

exceeded 100% (if there had not been a strengthening of ICU beds at the onset of the pandemic), suggesting in any case a substantial overburden on the healthcare system.

The highest percentage bed ICU occupancy varied from 15.75% (Calabria) to 270% (Valle d'Aosta); maximum ICU beds occupancy rate ranged from 1.18 (Calabria) to 21.48 per 100,000 inhabitants (Valle d'Aosta) (Table 2). The correlation with the delay was weak (0.38; 95%CI 0.08–0.68) (Fig 5, Panel A) for the highest percentage of ICU occupancy and moderate (0.50; 95%CI 0.21–0.79) for the maximum ICU beds occupancy rate (Fig 5, Panel B). It is of note that these two ICU measures were strongly correlated (0.95, data not shown).

Finally, COVID-19 mortality rate during the study period ranged in different regions from 0.05 to 1.5 per 1,000 inhabitants and was correlated at regional level with delay (correlation coefficient 0.70; 95%CI 0.49–0.92) (Fig 5, Panel C).

## Discussion

In Italy, strong anti-contagion interventions were implemented at national level in March 2020 when COVID-19 cumulative incidence reached high levels in some northern regions of the country, where the healthcare system was already overstressed by the increasing demand for intensive care. Our retrospective analysis shows that regions for which the lockdown was

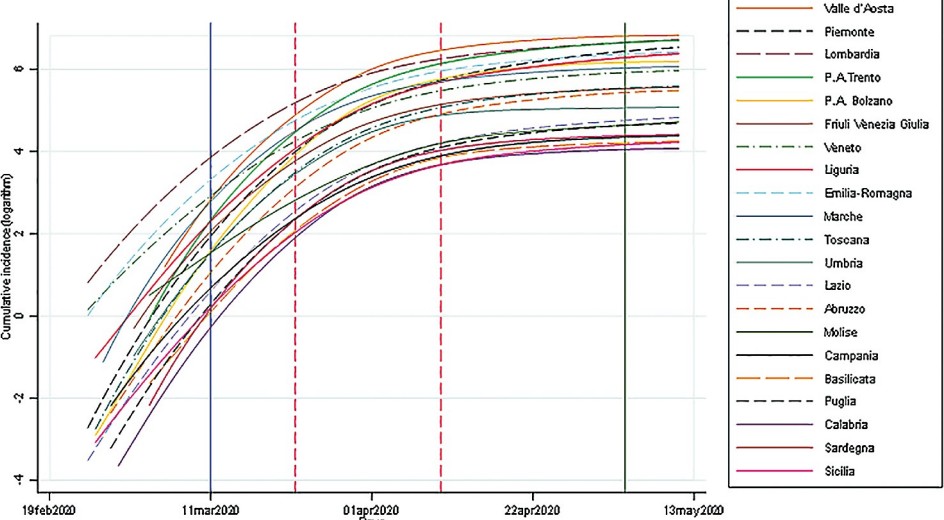

**Fig 1. Cumulative incidence of Covid-19 by region in Italy, February 19- May 13, 2020.** The vertical blue line is the National lockdown declaration; the green line is the end of our observation period; the red dash lines show the time range in which the number of new cases started to decrease.

enacted earlier, when the cumulative incidence was still low, had lower incidence and mortality during the first wave of the epidemic, and experienced a lower burden on the healthcare system as indicated by a manageable demand for intensive care unit beds.

Further, in this study, evaluating the impact of the first Covid-19 epidemic wave in Italy, we observed it was needed on average, 21 days (range 11–32), regardless of the delay in their implementation, from National lockdown to NPMs effectiveness (i.e., a decrease on the incidence of diagnosed cases). This result is an agreement with a recent study evaluating the change of reproduction number (R) that finding that national lockdown of March 11 brought R below 1 in most Italian regions and provinces within 2 weeks [13].

Previous studies have shown that the reproduction numbers of the epidemics were similar in all Italian regions during the first decade of March 2020 [14]. Accordingly, our analysis also found that all regions had similar epidemic trends during this period, suggesting a potential for similar growth of the epidemic in absence of NPIs. Thus, this study provides empirical support to previous modelling studies, suggesting that the timing of implementation of control measures is critical to achieve effective control of the epidemic—at least during its first wave. For example, Dehning et al. [15] estimated that a five-day delay in implementation of strict control measures in Germany would have resulted in a three-fold difference in cumulative cases. Similarly, Pei et al. [16] estimated that 61.6% of SARS-Cov2 infections reported in the USA as of May 3, 2020 could have been avoided if control measures had been implemented one week earlier. Moreover, in China, the size of the epidemic was limited more efficiently in cities that implemented control measures in the first week of their outbreaks compared with cities that started control later [17].

This work has several limitations that should be taken into account. First, the data used was aggregated the regional reported daily new cases of confirmed SARS-Cov-2 infections. Such data does not permit a more in depth analysis which would allow to standardize the CI and the other indicators here, used at least for age and sex population structure of each region. Furthermore, people microbiologically diagnosed with SARS-Cov-2, using RT-PCR in rhino-pharyngeal oral swab samples, are only part of the real infections occurred and this proportion of

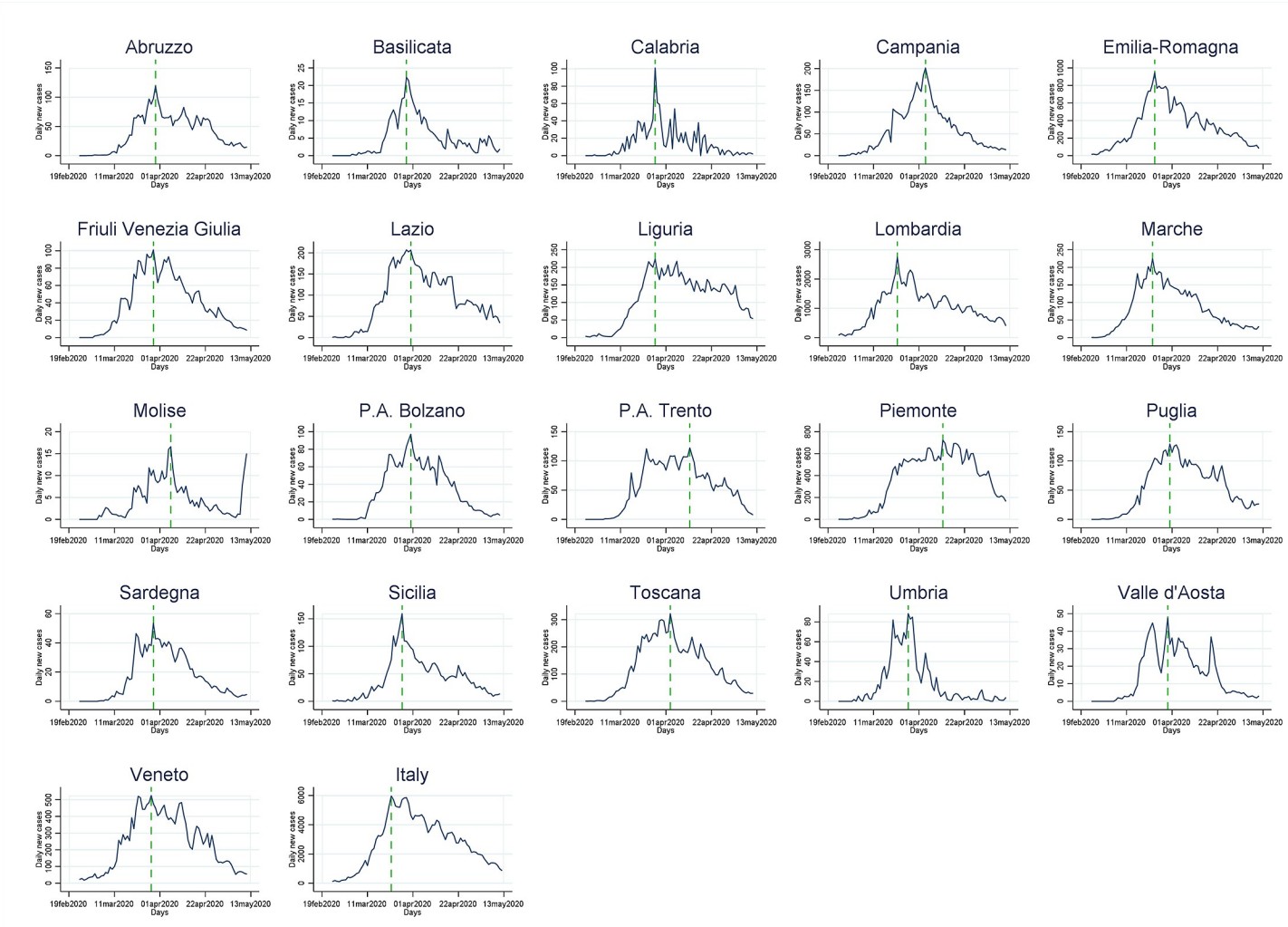

**Fig 2. Daily reported new cases of COVID-19 by region and overall Italy, February 19-May 13, 2020.** The vertical green line is the time when new cases started to decrease.

people diagnosed could vary by Region. A National serological survey evaluating the presence of IgG antibodies to SARS-Cov-2 estimated that, overall, around 1 out of 6 persons infected with SARS-Cov-2 were reported to the surveillance system [18], suggesting that the real impact measured by this data is limited. However, this data was substantially similar among the regions and thus not strongly biased with respect at the objective of this analysis. Analogous results, in terms of representativeness were also found in another study comparing the excess mortality observed in the period February-April in the Italian regions with deaths associated to Covid-19 reported to the Italian Surveillance system of Covid-19 [19]. Moreover, we did not considered in the analysis factors such as territorial conformation, level of urban and industrial development, territorial interconnections, etc. [20] who may influence the course of the epidemic and may vary among regions.

In conclusion, the present study highlight that early timing of implementation of containment measures on Covid-19 epidemic may have had a relevant impact on reducing the outbreak magnitude, on playing less pressure on ICU and on impacting the mortality associated

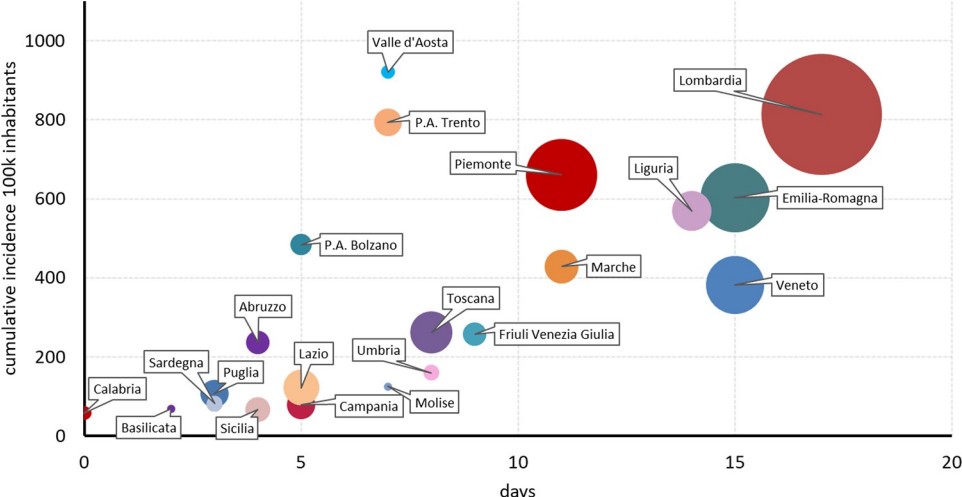

**Fig 3. Delay of lockdown declaration and cumulative incidence of Covid-19 at May 11<sup>th</sup> by region-Italy.** The horizontal axis is the number of days between the date on which each the region reached the cumulative incidence equal to the lowest regional CI at time of lockdown and the date of lockdown; the vertical is the cumulative incidence of Covid-19 (May 11<sup>th</sup>) per 100k inhabitants; the size of bubble is proportional to the total number of diagnosed positives (May, 11<sup>th</sup>).

to Covid-19 in Italian regions that where less impacted at the start of the epidemics. Interventions that may slow the spread of COVID 19, limit associated mortality and reduce its impact on the healthcare system are essentially based on limiting human mobility and reducing human activity including social and economic activity [21]. It is no surprise that these interventions have a dramatic socioeconomic impact [22] and that populations and governments

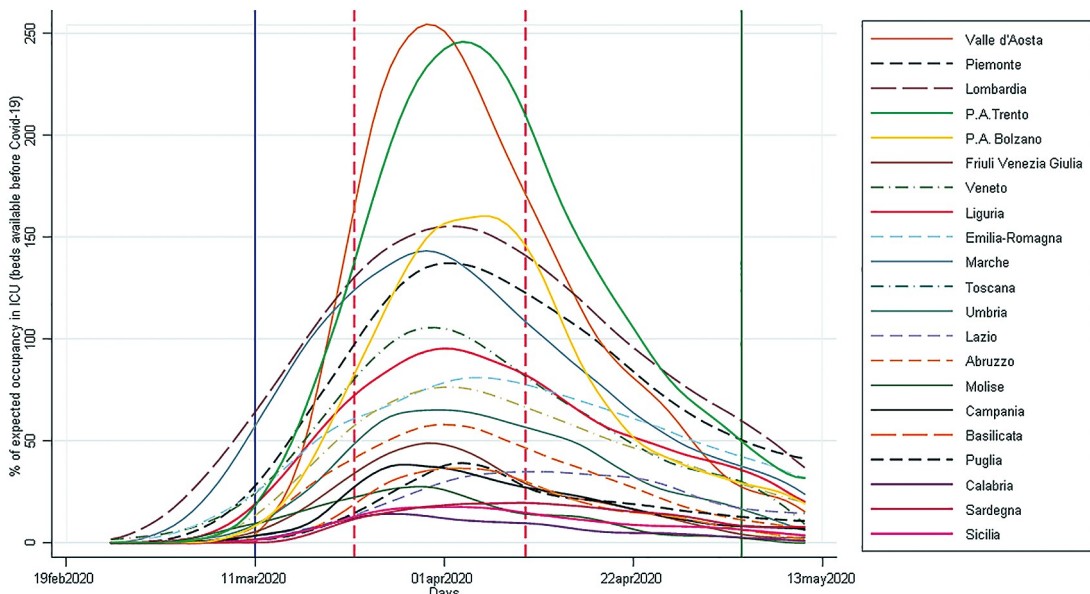

**Fig 4. Percentage of Intensive Care Unit (ICU) bed occupancy by region-Italy, February 19- May 13, 2020.** The vertical blue line is the National lockdown declaration; the green line is the end of our observation period; the red dash lines show the time range in which the number of new cases started to decrease.

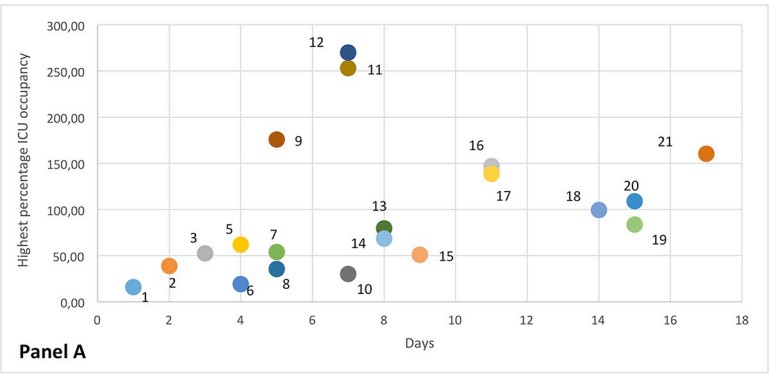

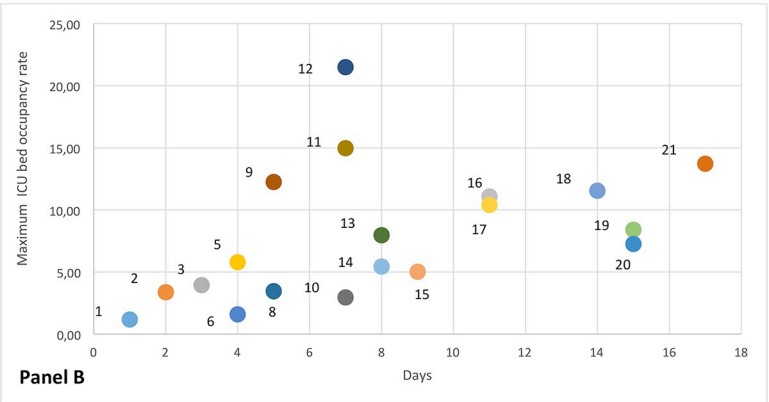

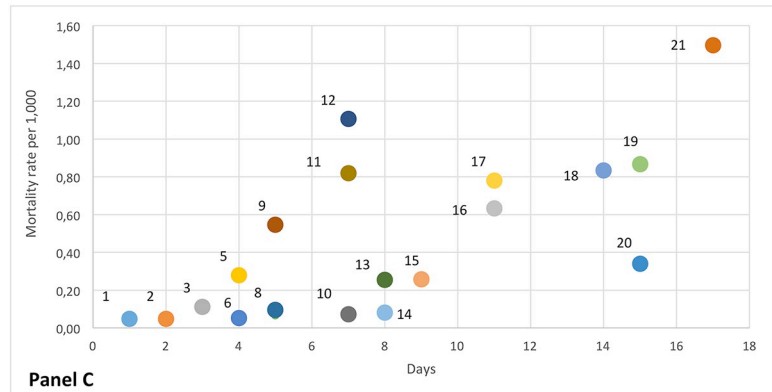

**Fig 5.** Scatterplots of highest percentage bed ICU occupancy (A), max ICU bed occupancy rate (B) and mortality rate at May 11th (C) versus delay of lockdown declaration. Numbers represent Regions as follows: Calabria (1), Basilicata (2), Puglia (3), Sardegna (4), Abruzzo (5), Sicilia (6), Campania (7), 8 Lazio (8), P.A. Bolzano (9), Molise (10), P.A. Trento (11), Valle d'Aosta (12), Toscana (13), Umbria (14), Friuli Venezia Giulia (15), Marche (16), Piemonte (17), Liguria (18), Emilia-Romagna (19), Veneto (20), Lombardia (21). The horizontal axis is the number of days between the date on which each the region reached the cumulative incidence equal to the lowest regional CI at time of lockdown and the date of lockdown.

may be hesitant in implementing these interventions. Nonetheless, our analysis supports the notion that non-pharmaceutical containment measures should be adopted very early during the course of Covid-19 epidemic in order to mitigate the impact on the healthcare system and to reduce related mortality.

## Author Contributions

**Conceptualization:** Enrico Girardi.

**Data curation:** Laura Timelli.

**Formal analysis:** Laura Timelli.

**Software:** Laura Timelli.

**Writing – original draft:** Laura Timelli, Enrico Girardi.

**Writing – review & editing:** Laura Timelli.

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
