## [Decision Letter · Decision Letter 0]

10 Dec 2020

PONE-D-20-34437

Effect of timing of implementation of containment measures on Covid-19 epidemic. The case of Italy.

PLOS ONE

Dear Dr. Timelli,

Thank you for submitting your manuscript to PLOS ONE. After careful consideration, we feel that it has merit but does not fully meet PLOS ONE’s publication criteria as it currently stands. Therefore, we invite you to submit a revised version of the manuscript that addresses the points raised during the review process.

Please address all the issues raised by the reviewers taking into consideration also all the comments/suggestions in editing the manuscript

We look forward to receiving your revised manuscript.

Kind regards,

Simone Lolli

Academic Editor

PLOS ONE

Journal Requirements:

2. In your Methods section, please give the sources for the following information:

i)    COVID-19 cases

ii)    ICU bed occupancy

Furthermore, this is a retrospective study; thus, we ask that you revise the text (especially, but no limited to, the aims and Conclusions) to avoid unsupported statements.

3. In your Data Availability statement, you have not specified where the minimal data set underlying the results described in your manuscript can be found; the link provided states data not found.

PLOS defines a study's minimal data set as the underlying data used to reach the conclusions drawn in the manuscript and any additional data required to replicate the reported study findings in their entirety. All PLOS journals require that the minimal data set be made fully available. For more information about our data policy, please see http://journals.plos.org/plosone/s/data-availability.

4. Please upload a new copy of Figure 2 as the detail is not clear. Please follow the link for more information: https://blogs.plos.org/plos/2019/06/looking-good-tips-for-creating-your-plos-figures-graphics/

5. Please include your tables as part of your main manuscript and remove the individual files. Please note that supplementary tables should be uploaded as separate "supporting information" files.

Reviewers' comments:

Reviewer's Responses to Questions

**Comments to the Author**

1. Is the manuscript technically sound, and do the data support the conclusions?

Reviewer #1: Yes

Reviewer #2: Yes

2. Has the statistical analysis been performed appropriately and rigorously? 

Reviewer #1: Yes

Reviewer #2: Yes

3. Have the authors made all data underlying the findings in their manuscript fully available?

Reviewer #1: Yes

Reviewer #2: Yes

4. Is the manuscript presented in an intelligible fashion and written in standard English?

Reviewer #1: Yes

Reviewer #2: Yes

5. Review Comments to the Author

Reviewer #1: The authors present a study that shows the impact of the timing of implementation of non-pharmaceutical containment measures (NPMs) on Covid-19 epidemic. They focus on the italian national lockdown, which started on March 11th 2020 and ended on May 4th 2020. The impact of NPMs on healthcare system has been evaluated in 21 italian local administrations (19 regions+2 autonomous provinces). Three indices have been used in order to assess this impact: cumulative incidence (CI), intensive care units occupancy (percentage on the total amount of ICUs), mortality rate. Timing of implementation of NPMs has been calculated as the number of days between the date on which each region reached a CI equal to 1.0 per 100k inhabitants, and the date of National Lockdown (it has been called delay). In the following lines some comments about the manuscript are reported:

1) The impact of the delay on healthcare system in terms of ICU occupancy and mortality rates is not sufficiently assessed. Two scatterplots (as Fig. 3) should be added: one between delay and highest ICU occupancy, the other one between delay and mortality rates (at May 11th). Related correlations should also be reported.

2) Looking at the data on Table 2, correlation between delay and mortality rate is about 0.7, which is close to the correlation between delay and CI. On the other hand, correlation between delay and highest percentage of ICU occupancy is lower (0.38). This probably means that the percentage of ICU occupancy is not the right statistic, and it should be replaced by the number of ICU beds occupied per 100k inhabitants, if these data are available. Percentage in fact is heavily affected by the total amount ICU beds, which is more closely linked to the quality level of the healthcare system than the spread of the epidemic.

3) NPMs effectiveness can be seen after 10 – 31 days (Fig. 1, 2 and 4), regardless of the delay in their implementation. The authors should put a little bit more emphasis on this result, in the abstract and in the Results section.

4) Figure 2 has a very low pixel resolution, it is almost unreadable. Moreover, daily cases have too many fluctuations: it would be better to replace them with weekly averages or moving averages.

5) References are ok.

Reviewer #2: This interesting paper explores how the timing of implementation of NPMs (or NPIs) impacted the cumulative incidence, ICU bed occupancy and mortality related to COVID-19 in Italy. I found the work clear and sufficiently explicative, providing support to modelling studies that suggested that the timing of implementation of control measures is critical to achieving effective control of the epidemic.

Among the strengths, I would mention the clarity of the aims and among the limitations, I would certainly agree that the quality of the data, aggregated, did not allow for in-depth analysis such as age and sex standardization.

The manuscript is very linear and coherent. Moreover, it is technically sound, and the data support the conclusions. Moreover, general conclusions are acceptable and in line with recent literature reports. I particularly appreciated the mention of the data coming from the Italian sero-survey and the report on excess mortality in the limitations section, as it gives a realistic picture of the reality of the pandemic in Italy while acknowledging the struggle of gathering complete data on new cases and deaths.

I believe the statistical analysis has been performed appropriately and rigorously. However, I would add a few details in the Methods/statistical analysis section, both technical, such as the retrospective nature of the study, and on aspects such as the specific software used to perform the analysis,

In Table 1 I would better explain the case-based measures, which do not seem clear.

The authors made all data underlying the findings in their manuscript fully available.

The manuscript is presented in an intelligible fashion and written in standard English, however, I would proofread it as a few sentences might be less clear if some typos are not corrected (e.g. figure 3 “shows a scatterplot by Region of the relationship of CI two months later the date of lockdown”, I believe it should be 2 months “after”), as well as some past tense forms or the s for the third person form of some verbs.

The aims, the discussion and the conclusions are coherent and sound, and ultimately support the adoption of early containment measures in order to counteract the epidemic surge and control its magnitude.

6. PLOS authors have the option to publish the peer review history of their article (what does this mean?). If published, this will include your full peer review and any attached files.

Reviewer #1: No

Reviewer #2: No

---

## [Author Response · Author response to Decision Letter 0]

30 Dec 2020

Answer: we checked /changed the file naming according to PLOS ONE's style requirements 

2. In your Methods section, please give the sources for the following information:

i) COVID-19 cases

Answer: we inserted the web link to the data as reported below:

https://github.com/pcm-dpc/COVID-19 (download 01/07/2020)

ii) ICU bed occupancy

Answer: we inserted the web link to the data as reported below: https://www.trovanorme.salute.gov.it/norme/renderNormsanPdf?anno=2020&codLeg=74348&parte=1%20&serie=null

Furthermore, this is a retrospective study; thus, we ask that you revise the text (especially, but no limited to, the aims and Conclusions) to avoid unsupported statements.

Answer: text was revised to take into account of this comment

 3. In your Data Availability statement, you have not specified where the minimal data set underlying the results described in your manuscript can be found; the link provided states data not found.

 Answer: as reported above, used data are available at the links provided:

Covid-19 cases � https://github.com/pcm-dpc/COVID-19 (downloaded 01/07/2020)

ICU bed � https://www.trovanorme.salute.gov.it/norme/renderNormsanPdf?anno=2020&codLeg=74348&parte=1%20&serie=null (Table 1)

Regional population at 01/01/2020 � www.demo.istat.it

4. Please upload a new copy of Figure 2 as the detail is not clear. 

 Answer: The new figure 2 is now provided with high pixel resolution

5. Please include your tables as part of your main manuscript and remove the individual files. Please note that supplementary tables should be uploaded as separate "supporting information" files.

 Answer: we included the tables as part of the manuscript.

Reviewer #1: The authors present a study that shows the impact of the timing of implementation of non-pharmaceutical containment measures (NPMs) on Covid-19 epidemic. They focus on the italian national lockdown, which started on March 11th 2020 and ended on May 4th 2020. The impact of NPMs on healthcare system has been evaluated in 21 italian local administrations (19 regions+2 autonomous provinces). Three indices have been used in order to assess this impact: cumulative incidence (CI), intensive care units occupancy (percentage on the total amount of ICUs), mortality rate. Timing of implementation of NPMs has been calculated as the number of days between the date on which each region reached a CI equal to 1.0 per 100k inhabitants, and the date of National Lockdown (it has been called delay). In the following lines some comments about the manuscript are reported:

1) The impact of the delay on healthcare system in terms of ICU occupancy and mortality rates is not sufficiently assessed. Two scatterplots (as Fig. 3) should be added: one between delay and highest ICU occupancy, the other one between delay and mortality rates (at May 11th). Related correlations should also be reported. 

Answer: we followed your suggestion and added two scatterplots, one between delay and highest ICU occupancy, the other one between delay and mortality rates (at May 11th) and relative correlations. Following also your comment number 2, we added the scatterplot between delay and the number of ICU beds occupied per 100k inhabitants (hereinafter referred to as maximum ICU beds occupancy rate) and relative correlation. We included in the revised version another figure (Figure 5) combining these new scatterplots in three panels. Also Table 1 was updated adding a column with maximum ICU beds occupancy rate. We also revised methods, results and discussion sections to take into account the new results.

2) Looking at the data on Table 2, correlation between delay and mortality rate is about 0.7, which is close to the correlation between delay and CI. On the other hand, correlation between delay and highest percentage of ICU occupancy is lower (0.38). This probably means that the percentage of ICU occupancy is not the right statistic, and it should be replaced by the number of ICU beds occupied per 100k inhabitants, if these data are available. Percentage in fact is heavily affected by the total amount ICU beds, which is more closely linked to the quality level of the healthcare system than the spread of the epidemic. 

Answer: Thanks for the advice. We calculated also the maximum ICU bed occupancy rate and relative correlation. We decided to maintain also the previous statistic as a measure both of quality level of the healthcare system and the impact of the spread on HS. It is of note that the correlation between the highest percentage of ICU occupancy maximum ICU bed occupancy rate is high (0.95). Below we show the scatterplot. We also revised methods, results and discussion sections to take into account the new results.

3) NPMs effectiveness can be seen after 10 – 31 days (Fig. 1, 2 and 4), regardless of the delay in their implementation. The authors should put a little bit more emphasis on this result, in the abstract and in the Results section. 

Answer: thanks for the suggestion. This is now highlighted in the revised version.

4) Figure 2 has a very low pixel resolution, it is almost unreadable. Moreover, daily cases have too many fluctuations: it would be better to replace them with weekly averages or moving averages. 

Answer: The new figure 2 is now with high pixel resolution; we also used weekly moving average, as suggested.

5) References are ok.

Answer: Thanks for the comment.

Reviewer #2: This interesting paper explores how the timing of implementation of NPMs (or NPIs) impacted the cumulative incidence, ICU bed occupancy and mortality related to COVID-19 in Italy. I found the work clear and sufficiently explicative, providing support to modelling studies that suggested that the timing of implementation of control measures is critical to achieving effective control of the epidemic.

Among the strengths, I would mention the clarity of the aims and among the limitations, I would certainly agree that the quality of the data, aggregated, did not allow for in-depth analysis such as age and sex standardization.

The manuscript is very linear and coherent. Moreover, it is technically sound, and the data support the conclusions. Moreover, general conclusions are acceptable and in line with recent literature reports. I particularly appreciated the mention of the data coming from the Italian sero-survey and the report on excess mortality in the limitations section, as it gives a realistic picture of the reality of the pandemic in Italy while acknowledging the struggle of gathering complete data on new cases and deaths.

I believe the statistical analysis has been performed appropriately and rigorously. However, I would add a few details in the Methods/statistical analysis section, both technical, such as the retrospective nature of the study, and on aspects such as the specific software used to perform the analysis

Answer: we thank the reviewer for these positive comments; following your suggestions, we specified that the study is retrospective and reported the software used for the analysis. 

In Table 1 I would better explain the case-based measures, which do not seem clear. 

Answer: we better specified the case-based measures, as suggested.

The authors made all data underlying the findings in their manuscript fully available.

The manuscript is presented in an intelligible fashion and written in standard English, however, I would proofread it as a few sentences might be less clear if some typos are not corrected (e.g. figure 3 “shows a scatterplot by Region of the relationship of CI two months later the date of lockdown”, I believe it should be 2 months “after”), as well as some past tense forms or the s for the third person form of some verbs.

Answer: the new version was further revised by a mother tongue person aiming to improve the language and to correct the typos.

The aims, the discussion and the conclusions are coherent and sound, and ultimately support the adoption of early containment measures in order to counteract the epidemic surge and control its magnitude.

Answer: thanks for the positive comment.

---

## [Decision Letter · Decision Letter 1]

6 Jan 2021

Effect of timing of implementation of containment measures on Covid-19 epidemic. The case of the first wave in Italy.

PONE-D-20-34437R1

Dear Dr. Timelli,

We’re pleased to inform you that your manuscript has been judged scientifically suitable for publication and will be formally accepted for publication once it meets all outstanding technical requirements.

Kind regards,

Simone Lolli

Academic Editor

PLOS ONE

Additional Editor Comments (optional):

I am happy to inform you that now the paper is ready for publication. The previously raised issues were addressed by the authors.

Reviewers' comments:

Reviewer's Responses to Questions

**Comments to the Author**

1. If the authors have adequately addressed your comments raised in a previous round of review and you feel that this manuscript is now acceptable for publication, you may indicate that here to bypass the “Comments to the Author” section, enter your conflict of interest statement in the “Confidential to Editor” section, and submit your "Accept" recommendation.

Reviewer #1: All comments have been addressed

Reviewer #2: All comments have been addressed

2. Is the manuscript technically sound, and do the data support the conclusions?

Reviewer #1: (No Response)

Reviewer #2: (No Response)

3. Has the statistical analysis been performed appropriately and rigorously? 

Reviewer #1: (No Response)

Reviewer #2: (No Response)

4. Have the authors made all data underlying the findings in their manuscript fully available?

Reviewer #1: (No Response)

Reviewer #2: (No Response)

5. Is the manuscript presented in an intelligible fashion and written in standard English?

Reviewer #1: (No Response)

Reviewer #2: (No Response)

6. Review Comments to the Author

Reviewer #1: (No Response)

Reviewer #2: (No Response)

7. PLOS authors have the option to publish the peer review history of their article (what does this mean?). If published, this will include your full peer review and any attached files.

Reviewer #1: No

Reviewer #2: No

---

## [Editor Report · Acceptance letter]

21 Jan 2021

PONE-D-20-34437R1 

Effect of timing of implementation of containment measures on Covid-19 epidemic. The case of the first wave in Italy. 

Dear Dr. Timelli:

I'm pleased to inform you that your manuscript has been deemed suitable for publication in PLOS ONE. Congratulations! Your manuscript is now with our production department. 

Kind regards, 

on behalf of

Dr. Simone Lolli 

Academic Editor

PLOS ONE